# FEATURE INTEGRATION AND GROUP TRANSFORMERS FOR ACTION PROPOSAL GENERATION

## ABSTRACT

The task of temporal action proposal generation (TAPG) aims to provide high-quality video segments, *i.e.*, proposals that potentially contain action events. The performance of tackling the TAPG task heavily depends on two key issues, *feature representation* and *scoring mechanism*. To simultaneously take account of both aspects, we introduce an attention-based model, termed as FITS, to address the issues for retrieving high-quality proposals. We first propose a novel Feature-Integration (FI) module to seamlessly fuse two-stream features concerning their interaction to yield a robust video segment representation. We then design a group of Transformer-driven Scorers (TS) to gain the temporal contextual supports over the representations for estimating the starting or ending boundary of an action event. Unlike most previous work to estimate action boundaries without considering the long-range temporal neighborhood, the proposed action-boundary co-estimation mechanism in TS leverages the bi-directional contextual supports for such boundary estimation, which shows the advantage of removing several false-positive boundary predictions. We conduct experiments on two challenging datasets, ActivityNet-1.3 and THUMOS-14. The experimental results demonstrate that the proposed FITS model consistently outperforms state-of-the-art TAPG methods.

## 1 INTRODUCTION

Owing to the fast development of digital cameras and online video services, the rapid growth of video sequences encourages the research of video content analysis. The applications of interest include video summarization (Yao et al., 2015; 2016), captioning (Chen et al., 2019a; Chen & Jiang, 2019), grounding (Chen et al., 2019b), and temporal action detection (Gao et al., 2019; Zhang et al., 2019). The temporal action detection task is an important topic related to several video content analysis methods, and it aims to detect the human-action instances within the untrimmed long video sequences. Like the image object detection task, the temporal action detection can be separated into a temporal action proposal generation (TAPG) stage and an action classification stage. Recent studies (Escorcia et al., 2016; Buch et al., 2017b; Lin et al., 2018; Liu et al., 2019; Lin et al., 2019; 2020) demonstrate that the way to pursue the proposal quality clearly improves the performance of two-stage temporal action detectors. To this end, a temporal action proposal generator is demanded to use a limited number of proposals for capturing the ground-truth action instances in a high recall rate, hence reducing the burden of the succeeding action classification stage.

One popular way to tackle the TAPG task is to generate the proposals via the estimations of boundary and actionness probabilities. The boundary probability is usually factorized as the starting and ending for an action instance. Rather than directly estimating the actionness boundary as the existing methods, we leverage the actionness estimation and the additional *background* estimation in a bi-directional temporal manner to co-estimate the action boundaries. The background means existing no actions. This sort of boundary estimation derived from the observation that the features for describing the long-time actionness/background are more consistent along the temporal dimension than the short-time starting/ending. Therefore, estimating the boundary with the actionness and background features allows us to estimate the proposal boundaries of much less false-positive, hence obtaining the high-quality proposal candidates for further scoring. In practice, we estimate an action starting boundary as the time of descending background with simultaneous ascending actionness. In contrast, an action ending boundary occurs with the ascending background with descending actionness. Figure 1 illustrates our action-boundary co-estimation mechanism.

Figure 1: The proposed action-boundary co-estimation mechanism. Our transformer-driven scorers module estimates four boundary estimations of forward-actioness, backward-actioness, forward-background, and backward-background. The backward-actioness and forward-background co-estimate the action starting, and the forward-actioness and backward-background co-estimate the action ending. The transformer-style units enable these temporal estimations to collect the temporal contextual supports over each inputted representation. The right-most figures show the estimations of action starting and action ending without (top) and with (bottom) the TS co-estimation. The proposed action-boundary co-estimation within our TS module is able to reduce more false-positive predictions.

This paper introduces an effective temporal action proposal generator, *i.e.*, FITS, which aims to provide the action proposals, that precisely and exhaustively cover the human-action instances. By considering the two essential TAPG issues, namely the feature representation and the scoring mechanism, and the above-mentioned action-boundary co-estimation, our attention-based FITS model comprises *Feature Integration (FI)* module and *Transformer-driven Scorers (TS)* module for dealing with these considerations. Precisely, our FI module enhances the common TAPG two-stream features (Simonyan & Zisserman, 2014; Wang et al., 2016; Xiong et al., 2016) by concerning the feature interaction. The previous TAPG methods usually directly concatenate the appearance stream and motion stream features for usage. In contrast, we were inspired by the non-local attention mechanisms (Wang et al., 2018; Hsieh et al., 2019) to extend such a long-range attention mechanism for integrating the two-stream features. As a result, our experiments show the robustness of the integrated features by reducing their mutual feature discrepancies. More importantly, to score the temporal action proposals for discriminating high-quality ones, we devise a novel transformer-driven scoring mechanism. The TS mechanism leverages the temporal contextual supports over the feature representations to obtain the self-attended representations and then associates these self-attended representations to co-estimate the action boundaries. The experiments show the retrieved action proposals containing much less false-positive ones. Figure 2 overviews our temporal action proposal model, termed as FITS network.

To sum up, our main contributions include *i*) We introduce the novel *feature integration* module to integrate the two-stream features by reducing their feature discrepancies via non-local-style attention and obtaining robust representation. *ii*) We devise the novel *transformer-driven scorers* module to co-estimate the transformer-driven self-attended representations, which leverage long-range temporal contextual supports. Hence, we are able to retrieve high-quality temporal action proposals. *iii*) The extensive experiments demonstrate that the proposed FITS model achieves significantly better performance than current state-of-the-art TAPG methods.

## 2 RELATED WORK

**Feature Representation.**    As a de facto trend, instead of using the handcrafted features, the neural-network-based features are widely employed for addressing the action classification task. These popular neural network approaches include the two-stream networks (Simonyan & Zisserman, 2014; Feichtenhofer et al., 2016; Wang et al., 2016), which separately represent the appearance feature and the motion feature, and 3D networks (Tran et al., 2015; Carreira & Zisserman, 2017; Qiu et al., 2017; Xu et al., 2017), which directly represent a video sequence as the spatio-temporal feature. In this paper, we use the action recognition model (Wang et al., 2016; Xiong et al., 2016) to extract two-stream features for representing each untrimmed video sequence.

**Attention Mechanism.** The attention mechanism is the process of selectively focusing on a few relevant things in comparison with everything. Fields like natural language processing and computer vision, broadly leverage such an attention mechanism. For instances, Bahdanau et al. (2015) enable their model to focus on searching a group of related words from the input sentences for predicting the target words, Xu et al. (2015) introduce the soft and hard attention to generate image captions, and LFB (Wu et al., 2019) introduces long-term feature banks to analyze videos, Hsieh et al. (2019) employ non-local attention (Wang et al., 2018) to capture long-range dependencies and channel attention (Hu et al., 2018) to re-weight the channel-level feature maps. Since Vaswani et al. (2017) introduce a self-attention mechanism, called *Transformer*, for exploring the intra-attention and inter-attention to address the machine translation task, the Transformer-based models show their advantages to tackle various tasks, such as object detection (Carion et al., 2020), action recognition (Girdhar et al., 2019), image generation (Parmar et al., 2018), and image captioning (Cornia et al., 2020). We notice that the work of Girdhar et al. (2019) that classifies the human actions by first use the RPN (Ren et al., 2015) to localize the human body parts as several frame crops, and then encodes these crops via Transformers for the subsequent classification. In contrast, a TAPG task discriminates whether a given video segment, composed of several video frames, covering any action event. Our model extends the merits of these attention efforts for constructing a robust action proposal generator.

**Temporal Action Proposal Generation.** We categorize the TAPG methods into *Anchor-based* (Gao et al., 2017; Heilbron et al., 2016; Shou et al., 2016) and *Probability-based* (Lin et al., 2020; 2019; 2018; Zhao et al., 2017). The former focuses on designing several multi-scale anchor boxes to cover action instances, while the latter estimates the temporal location probabilities of the action instances. Besides, some methods (Gao et al., 2018; Liu et al., 2019; Gao et al., 2020) also explore the way to integrate the above-mentioned two categories for precisely localizing the temporal boundaries. In anchor-based methods, the S-CNN (Shou et al., 2016) and Heilbron et al. (2016) respectively evaluate anchors via C3D network and sparse learning, and TURN (Gao et al., 2017) suggests regressing the temporal boundaries of action instances. The probability-based work, TAG (Zhao et al., 2017), generates action proposals via a temporal watershed algorithm to merge contiguous temporal locations of high actioness probabilities. BSN (Lin et al., 2018) generates proposals as well as their confidence by formulating the probabilities of boundaries and actioness. BMN (Lin et al., 2019) proposes a boundary-matching mechanism to evaluate the confidence among densely distributed proposals. DBG (Lin et al., 2020) uses the maps of dense boundary confidence and completeness to further score boundaries for all action proposals. BC-GNN (Bai et al., 2020) employs a Graph Neural Network to model the relationship between the actioness and boundaries. Zhao et al. (2020) propose loss terms to regularize the feature consistency between the actioness and boundaries.

In sum, the anchor-based methods focus on the anchor-box design and usually lack the flexible temporal boundaries for various action instances. The existing probability-based methods may generate the action proposals, but merely rely on the actioness (Zhao et al., 2017) or the boundary information (Lin et al., 2019), or lack the association between the actioness and the boundaries of action instances (Lin et al., 2020; 2018). By contrast, we introduce the Transformer-driven scorers module to score proposals by explicitly associating the self-attended representations during scorers training. Combining with our feature integration module for retrieving the robust feature representation, the experiments show that our model outperforms the existing leading TAPG methods.

## 3 METHOD

**TAPG Formulation.** Given a video sequence $\mathcal{X} = \{x_n\}_{n=1}^N$ of $N$ frames, we assume it includes $I$ action instances. The TAPG task aims at generating a proper set of action proposals that can be used to detect the underlying human actions in $\mathcal{X}$. We denote an action proposal as $p = (s, e)$ to indicate that the proposal $p$ starts at the $s$th frame and ends at the $e$th frame of $\mathcal{X}$. Analogous to the object proposals for detection, action proposals are generic and class agnostic. Let the $I$ ground-truth actions of $\mathcal{X}$ be $\mathcal{Y} = \{(s^*, e^*)_i = (s_i^*, e_i^*)\}_{i=1}^I$. An action proposal $(s, e)$ is said to be matched to some ground-truth action $(s^*, e^*)$ if their time-interval IoU (in terms of frames) is greater than a specified threshold $\tau$. Considering a set $\mathcal{P}$ of proposals, its goodness to $\mathcal{X}$ can be explicitly measured by the number of matched action proposals.

Figure 2: The architecture of our FITS temporal action proposal generation model. Our model is composed of a *Feature Integration* module, in charge of integrating the two-stream representations, and a *Transformer-driven Scorers* module, in charge of extracting the temporal contextual supports within representation and then associating these representations to co-estimate the action boundaries.

## 3.1 FEATURE INTEGRATION

We decompose each video sequence $\mathcal{X}$ into a set of, say, $T$ consecutive video segments, denoted as $\mathcal{V} = \{\mathbf{v}_t\}_{t=1}^T$, where $\mathbf{v}$ is called a *snippet*. Each snippet $\mathbf{v}$ is then represented with two-stream features. To account for videos of various lengths, we adopt the setting in BSN (Lin et al., 2018) to sample single-stream features over the temporal (frame) dimension to consistently obtain $T$ snippets per video sequence. Precisely, each video sequence $\mathcal{X}$ is represented by an appearance feature tensor $A \in \mathbb{R}^{C \times T}$ and a motion feature tensor $M \in \mathbb{R}^{C \times T}$. To enhance the representation power of each snippet, we enrich the representation by three-step operations: *co-attention*, *mutual-excitation*, and *aggregation*. Before we describe the steps, we first define a basic convolutional layer $\phi$ by

$$\phi(X; f, o) = \mathrm{ReLU}(\mathbf{W} * X + \mathbf{b}),  \tag{1}$$

where $X$, $f$, $o$ respectively denotes the input feature, the filter size, and the number of the output filters. ReLU, $*$, $\mathbf{W}$, $\mathbf{b}$ means the activation, convolution operation, the weights and bias of $\phi$.

**Co-attention.** We consider the non-local function (Hsieh et al., 2019; Wang et al., 2018) to explore the snippet-level correlations between the two-stream cues. Hence we enable our model to learn to emphasize the temporal correlations of human-action descriptions between the two-stream cues. Given a video sequence $\mathcal{X}$, its appearance tensor $A \in \mathbb{R}^{C \times T}$ and motion tensor $M \in \mathbb{R}^{C \times T}$, the proposed two-stream co-attention is achieved by carrying out the following feature re-adjustment:

$$\hat{A} = \Phi(A; M) \oplus A, \quad \hat{M} = \Phi(M; A) \oplus M,  \tag{2}$$

where $\hat{A}, \hat{M} \in \mathbb{R}^{C \times T}$, the non-local function $\Phi(\cdot)$ yields $T \times T$ snippet-level feature correlations for feature adjustment that *conditioning on the other feature*, and $\oplus$ denotes the element-wise addition.

**Mutual-excitation.** We further enhance the two-stream features by re-weighting over the *channel* dimension $C$. To this end, we introduce a mutual-excitation mechanism to re-weight each single-stream feature *concerning the channel-attention from the other stream*. A convolution layer $\phi$ of filter size $1 \times 1$ over the channel dimension is applied to re-organize the two-stream features. With the adjusted features, we construct a two-stream mutual-excitation function $\Psi$ to extract the channel attention vector from one stream to excite the feature of the other stream. That is, we have

$$\tilde{A} = \Psi(\phi(\hat{M}; 1 \times 1, C)) \odot \phi(\hat{A}; 1 \times 1, C), \quad \tilde{M} = \Psi(\phi(\hat{A}; 1 \times 1, C)) \odot \phi(\hat{M}; 1 \times 1, C),  \tag{3}$$

where $\tilde{A}, \tilde{M} \in \mathbb{R}^{C \times T}$, the mutual-excitation function $\Psi(\cdot)$ is generalized from squeeze-and-excitation (Hu et al., 2018) for describing channel-attention, and $\odot$ symbolizes the element-wise multiplication.

**Aggregation.** Inspired by the inception block (Szegedy et al., 2015; 2016), we further enrich the feature representation concerning the multi-scale temporal contexts before integrating the two-stream features. Given the two-stream features $\tilde{A}$ and $\tilde{M}$, we employ the convolution layer and max-pooling layer to respectively map each of the two-stream features in two different temporal contexts by

$$\bar{A}_1 = \phi(\tilde{A}; 1 \times 3, C'), \ \bar{A}_2 = \phi(\rho(\tilde{A}); 1 \times 3, C'), \ \bar{M}_1 = \phi(\tilde{M}; 1 \times 3, C'), \ \bar{M}_2 = \phi(\rho(\tilde{M}); 1 \times 3, C'),  \tag{4}$$

where $\bar{A}_1, \bar{A}_2, \bar{M}_1, \bar{M}_2 \in \mathbb{R}^{C' \times T}$, the max-pooling function $\rho(\cdot)$ using the filter of size $1 \times 3$. We then concatenate all features followed by another convolution layer to unify them. Formally, the aggregated feature, say $\mathbf{F}$, is generated by

$$\mathbf{F} = \phi(\bar{A}_1 \parallel \bar{A}_2 \parallel \bar{M}_1 \parallel \bar{M}_2; 1 \times 3 \times 4, C''), \tag{5}$$

where the notation $\parallel$ symbolizes the concatenation over an augmented last dimension, namely, $(\bar{A}_1 \parallel \bar{A}_2 \parallel \bar{M}_1 \parallel \bar{M}_2) \in \mathbb{R}^{C' \times T \times 4}$, and $\mathbf{F} \in \mathbb{R}^{C'' \times T}$ is squeezed over the augmented last dimension.

## 3.2 Transformer-driven Scorers

Our transformer-driven scorers comprises four encoder-decoder pairs. Both encoder and decoder are made of stacks of attention layers. Each encoder-decoder pair, *i.e.*, a Transformer unit, is in charge of extracting the inputted representation $\mathbf{F}$'s temporal contextual supports, in which the temporal contextual supports consider one single temporal direction with respect to the actioness or background. Hence, our encoder-decoder pairs generate the attended representations $\mathbf{F}^{\text{fa}}, \mathbf{F}^{\text{ba}}, \mathbf{F}^{\text{fb}}, \mathbf{F}^{\text{bb}}$ for forward-actioness, backward-actioness, forward-background, and backward-background, respectively. In Figure 2, the left part sketches our FITS model, and the right part details the architecture of one transformer units, *i.e.*, encoder-decoder. Precisely, we employ a one-layer eight-head Transformer with additional head-selection operation as one single encoder-decoder pair within our FITS model.

**Basic Units.** Since our model shares the same Transformer's basic units, *i.e.*, positional encoding, multi-head attention, add & norm, feed forward network, and masked multi-head attention, we refer the readers to Vaswani et al. (2017) for more details about these units. Here we briefly review the original Transformer architecture. The Transformer is a self-attention mechanism to compare each feature with all other features in the sequence. This paper employs such a self-attention mechanism to gain the temporal contextual supports from all other snippets for the snippet-level representation. The mechanism embeds the inputted feature to query $Q$, key $K$, and value $V$ via linear projections. The product of $Q$ and $K$ formulates the attention weights for adjusting the value $V$ as the output.

**Transformer Unit.** Each of our encoder-decoder pair takes a representation of $\mathbf{F}$ as input, and map it into query, key, and value. For a snippet of one action event as the query, the encoder-decoder pair then compares the relationship between the query and the keys of all other snippets to update the snippet's value. Intuitively, the self-attention process collects the contextual information from all other snippets to adjust the queried snippet's representation. Therefore, each snippet-level prediction actually considers the other snippets and hence smooths the prediction results, as shown in Figure 1. We now show the method to employ our encoder-decoder pairs. An attention function of a $\text{head}_i$ (Vaswani et al., 2017) is defined as:

$$\text{head}_i(Q, K, V) = \text{softmax}(\frac{QK^{\mathsf{T}}}{\sqrt{d}})V, \tag{6}$$

where $d$ denotes the normalization term. The $\text{head}$ attention function is the main component of the multi-head attention module and the masked multi-head attention module in the Transformer. Yet we use the additional head selection to fuse all heads within these modules. Our encoder and decoder empirically respectively take the representations of $\tilde{A} \parallel \tilde{M}$ and $\mathbf{F}$ as inputs. We use four encoder-decoder pairs to generate the attended representations $\mathbf{F}^{\text{fa}}, \mathbf{F}^{\text{ba}}, \mathbf{F}^{\text{fb}}, \mathbf{F}^{\text{bb}} \in \mathbb{R}^{C'' \times T}$. Note that the inputs for $\mathbf{F}^{\text{ba}}$ and $\mathbf{F}^{\text{bb}}$ are reversed over the temporal dimension concerning the backward temporal relation. For an attended representations $\mathbf{F}^i \in \{\mathbf{F}^{\text{fa}}, \mathbf{F}^{\text{ba}}, \mathbf{F}^{\text{fb}}, \mathbf{F}^{\text{bb}}\}$, we use (1) to predict its probability $\mathbf{P}^i = \phi(\mathbf{F}^i; 1 \times 3, 1) \in \mathbb{R}^{1 \times T}$. Hence, we can obtain the probabilities $\mathbf{P}^{\text{fa}}, \mathbf{P}^{\text{ba}}, \mathbf{P}^{\text{fb}}, \mathbf{P}^{\text{bb}}$.

**Head Selection.** Unlike Vaswani et al. (2017) to directly concatenate all heads' results with a subsequent linear projection, we present the head selection to make each neuron adaptive select its representation from the multiple heads. The head selection $\xi$ is defined as

$$\xi(\{\text{head}_i\}) = \phi(\sum(\text{scale}_i \times \text{head}_i); 1 \times 1, C''), \tag{7}$$

$$\text{scale}_i = \text{softmax}(\text{FC}(\text{GAP}(\sum \text{head}_i))), \tag{8}$$

where $\text{head}_i$ denotes the $i$th head result, $\sum$ denotes the element-wise summation, FC and GAP are fully-connection and global average pooling, respectively. Note that the $\text{softmax}$ in (8) operates over the head dimension rather than the channel or temporal dimension. The head selection is designed to adjust the channel scales for enhance the representation power as the SE-Net (Hu et al., 2018).

**Scorers.** Intuitively, a snippet $\mathbf{v}_i$ could be an action starting boundary if its previous snippet $\mathbf{v}_{i-1}$ is considered as a background, and its subsequent snippet $\mathbf{v}_{i+1}$ is likely as an action instance. On the other hand, a snippet $\mathbf{v}_i$ could be an action ending boundary if $\mathbf{v}_{i-1}$ is also as an action instance, and $\mathbf{v}_{i+1}$ is likely as a background. Bearing these observations in mind, we predict the probabilities of action starting $\mathrm{P}^{\mathrm{s}}$ and action ending $\mathrm{P}^{\mathrm{e}}$ over all snippets with two convolution layers by

$$\mathrm{P}^{\mathrm{s}} = \phi(\phi(\mathbf{F}^{\mathrm{ba}} \| \mathbf{F}^{\mathrm{fb}}; 1 \times 3, C); 1 \times 1, 1)\,, \tag{9}$$

$$\mathrm{P}^{\mathrm{e}} = \phi(\phi(\mathbf{F}^{\mathrm{fa}} \| \mathbf{F}^{\mathrm{bb}}; 1 \times 3, C); 1 \times 1, 1)\,, \tag{10}$$

where $\mathrm{P}^{\mathrm{s}}, \mathrm{P}^{\mathrm{e}} \in \mathbb{R}^{1 \times T}$, the notation $\|$ symbolizes the concatenation over the channel-dimension. Besides the snippet-level probabilities $\mathrm{P}^{\mathrm{s}}$ and $\mathrm{P}^{\mathrm{e}}$, we further consider the proposal-level probabilities. Given a proposal starting from the $i$th snippet to the $j$th snippet, we predict the action-covering probability $\mathrm{P}^{\mathrm{c}}$ and boundary-relation probability $\mathrm{P}^{\mathrm{se}}$ as

$$\mathrm{P}^{\mathrm{c}} = \phi(\mathbf{H}^{\mathrm{c}}; 1 \times 3 \times 3, 1)\,, \quad \mathrm{P}^{\mathrm{se}} = \phi(\mathbf{H}^{\mathrm{se}}; 1 \times 3 \times 3, 1)\,, \tag{11}$$

where $\mathrm{P}^{\mathrm{c}}, \mathrm{P}^{\mathrm{se}} \in \mathbb{R}^{1 \times T \times T}$. Each element $\mathbf{H}^{\mathrm{c}}_{i,j} \in \mathbf{H}^{\mathrm{c}}$ indicates the truncated probability vector from the $i$th bin to the $j$th bin, followed by ROI-Align (Ren et al., 2015), where the probability vector is $\phi(\mathbf{F}^{\mathrm{fa}} \| \mathbf{F}^{\mathrm{ba}}; 1 \times 3, 1)$. Each element $\mathbf{H}^{\mathrm{se}}_{i,j} \in \mathbf{H}^{\mathrm{se}}$ indicates two truncated intermediate representations of (9) and (10) from the $i$th snippet to the $j$th snippet, followed by element-wise addition, where the intermediate representations are the results of the first convolution layer of (9) and (10). Note that we calculate $\mathrm{P}^{\mathrm{c}}$ and $\mathrm{P}^{\mathrm{se}}$ by using the filter of size $3 \times 3$ to consider the neighboring proposals.

Given an action proposal starting from the $i$th snippet to the $j$th snippet, we empirically define its score $s_{ij}$ with the probabilities mentioned above as:

$$s_{ij} = \exp(\mathrm{P}^{\mathrm{s}}_i \cdot \mathrm{P}^{\mathrm{e}}_j) \times \mathrm{P}^{\mathrm{se}}_{ij} \times (\mathrm{P}^{\mathrm{c}}_{ij})^{\frac{1}{4}}\,, \tag{12}$$

where $\exp(\mathrm{P}^{\mathrm{s}}_i \cdot \mathrm{P}^{\mathrm{e}}_j)$, $\mathrm{P}^{\mathrm{se}}_{ij}$ and $(\mathrm{P}^{\mathrm{c}}_{ij})^{1/4}$ are introduced for respectively preferring high boundary confidence, high boundary correlation for a starting-ending pair, and high overlapping ratio between proposal and action-instance. In our implementation, we follow BSN (Lin et al., 2018) to collect candidate proposals by employing the probabilities of $\mathrm{P}^{\mathrm{s}}$ and $\mathrm{P}^{\mathrm{e}}$. Thus, each proposal is scored via (12) and followed by the soft non-maximum suppression for retrieving the top-scored proposals.

### 3.3 OPTIMIZATION

The overall loss function for training is formulated as a multi-task objective that comprises encoding loss ($\mathcal{L}_{enc}$) and scoring loss ($\mathcal{L}_{scr}$):

$$\mathcal{L} = \mathcal{L}_{enc} + \lambda\,\mathcal{L}_{scr}\,, \tag{13}$$

where $\mathcal{L}_{enc}$ is used for training the four probabilities $\mathrm{P}^{\mathrm{fa}}, \mathrm{P}^{\mathrm{ba}}, \mathrm{P}^{\mathrm{fb}}, \mathrm{P}^{\mathrm{bb}}$, and $\mathcal{L}_{scr}$ is designed for learning the remaining probabilities $\mathrm{P}^{\mathrm{s}}, \mathrm{P}^{\mathrm{e}}, \mathrm{P}^{\mathrm{c}}$, and $\mathrm{P}^{\mathrm{se}}$. The encoding loss $\mathcal{L}_{enc}$ encourages each Transformer to encode all its representations for predicting the probabilities of actioness or background, and the scoring loss $\mathcal{L}_{scr}$ encourages all Transformers to correlate their representations for ranking the proposals. Both the loss terms in (13) employ the binary logistic regression loss.

## 4 EXPERIMENTS

**Datasets and Metrics.** We conduct experiments on ActivityNet-1.3 (Heilbron et al., 2015) dataset and THUMOS-14 (Jiang et al., 2014) dataset. The ActivityNet-1.3 is a large-scale action understanding dataset, which is available for evaluating the tasks of proposal generation, action recognition, temporal detection, and dense captioning. There are 19,994 temporal annotated untrimmed videos comprising 200 action categories. The THUMOS-14 dataset contains 1,010 validation videos and 1,574 testing videos of 20 action categories. There are 200 validation videos, and 212 testing videos contain temporal action annotations. We use the validation set for training and use the testing set for evaluating. We use three metrics to evaluate the proposal quality. *i)* AR@AN, which evaluates the relation between Average Recall (AR), that calculated with multiple specified IoU thresholds, and Average Number of proposals (AN). *ii)* AUC, which denotes the area under the AR vs. AN curve.

Table 1: Comparison of the state-of-the-art methods on ActivityNet-1.3 validation and testing split and on THUMOS-14 testing split. Notation "*" indicates the model using non-two-stream features.

| Method | ActivityNet-1.3 | | | THUMOS-14 | | | | |
|---|---|---|---|---|---|---|---|---|
| | AR@100 (val) | AUC (val) | AUC (test) | AR@50 | AR@100 | AR@200 | AR@500 | AR@1000 |
| *GTAN (Long et al., 2019) | 74.80 | 67.10 | 67.40 | - | - | 54.30 | - | - |
| BMN (Lin et al., 2019) | 75.01 | 67.10 | 67.19 | 39.36 | 47.72 | 54.70 | 62.07 | 65.49 |
| RapNet (Gao et al., 2020) | 76.71 | 67.63 | 67.72 | 40.35 | 48.23 | 54.92 | 61.41 | 64.47 |
| DBG (Lin et al., 2020) | 76.65 | 68.23 | 68.57 | 37.32 | 46.67 | 54.50 | 62.21 | 66.40 |
| Zhao et al. (2020) | 75.27 | 66.51 | - | **44.23** | **50.67** | 55.74 | - | - |
| BC-GNN (Bai et al., 2020) | 76.73 | 68.05 | - | 40.50 | 49.60 | 56.33 | 62.80 | 66.57 |
| FITS | **77.59** | **69.36** | **70.00** | 40.32 | 49.53 | **57.55** | **65.34** | **69.56** |

**Label Assignment.** We adopt actioness $P^a$, action starting $P^s$, and action ending $P^e$ to estimate each frame's probabilities being an actioness, action-starting point, and action-ending point, respectively. Given a video sequence $\mathcal{X} = \{x_n\}_{n=1}^N$ of $N$ frames, we assume it includes $I$ action instances. Hence the ground-truth actions of $\mathcal{X}$ are $\mathcal{Y} = \{(s^*, e^*)_i = (s_i^*, e_i^*)\}_{i=1}^I$. We then define the ground-truth probabilities of $P^a$, $P^s$, and $P^e$ within a time span of $i$th action instance as

$$T_i^a = [s_i^*, e_i^*] , \quad T_i^s = [s_i^* - \delta_i, s_i^* + \delta_i] , \quad T_i^e = [e_i^* - \delta_i, e_i^* + \delta_i] , \quad (14)$$

where we shift the time span by $\delta_i = (e^* - s^*)/\eta$, and we set $\eta$ to 40 empirically. Given the $n$th frame, we represent the corresponding time span as $T_n = [n, n+1]$. We then use $T_n$ to calculate the overlap ratio with respect to $T_i^a$, $T_i^s$, and $T_i^e$ for actioness, action starting, and action ending, respectively. The overlap ratio serves as the ground-truth probability. The probabilities of actioness and background are complementary with a summation of 1. Note that the ground-truth actioness probabilities for both directions, i.e., $P^{fa}$ and $P^{ba}$, are the same. Analogously, the ground-truth background probabilities of $P^{fb}$ and $P^{bb}$ are the same.

We take two aspects to define proposal-level confidence scores. First, we determine the action-covering ratio as the probability $P^c$ for each action proposal $(s, e)$ by calculating the time-interval IoU with all action instances $I$. The action instance of the highest IoU value is regarded as the ground-truth action-covering score of the proposal. That is, a time-interval IoU determines the coverage between an action proposal and action instances within a time interval of $[s, e]$. We then employ the aspect of the proposal's boundary to define the confidence score. Higher boundary probability potentially denotes an action starting at the $s$th frame and ending at the $e$th frame. Specifically, we obtain the ground-truth boundary-relation probabilities of $P^{se}$ for all proposals by applying the outer product between ground-truth probabilities of $P^s$ and $P^e$.

## 4.1 COMPARISON WITH STATE-OF-THE-ARTS

Table 1 summarizes the comparison of our approach against state-of-the-art TAPG methods. Our model significantly outperforms other TAPG methods on all metrics of both datasets. Specially, we improve AR@100 and AUC of ActivityNet-1.3 validation split by $0.86\%$ and $1.13\%$, respectively. On the ActivityNet-1.3 testing split evaluating on the official server, we further improve AUC by $1.43\%$. On the THUMOS-14 testing splits, though the very recent method Zhao et al. (2020) shows the outstanding ranking results and hence surpasses FITS at $AR@50$ and $AR@100$, our FITS model surpasses all other methods with a sufficient number of available proposals. This comparison also shows that our FITS is able to retain more high-quality proposals compared to other TAPG methods.

## 4.2 ABLATION STUDY

We conduct a detailed ablation study on ActivityNet-1.3 validation split to realize the importance per component of our FITS model. Table 2 summarizes the ablation study of the feature integration module and transformer-driven scorers module, and please notice we assume a single module in this ablation equips with all the components of the other module. Table 3 summarizes the ablation study of the configurations in the transformer-driven scorers module.

**Feature Integration.** The left part in Table 2 compares the components within the feature integration module. The first row, i.e., baseline-FI, serves as the baseline that concatenates the two-stream

Table 2: Ablation study on ActivityNet-1.3 validation split. The meanings of abbreviations are CA: co-attention; ME: mutual-excitation; AG: aggregation; PG: performance gain on AUC; baseline-FI: concatenating the two-stream features; baseline-TS: estimating action boundaries as BSN and BMN.

| Component | | | Feature Integration | | | | | | Component | | | Transformer-driven Scorers | | | | | |
|---|---|---|---|---|---|---|---|---|---|---|---|---|---|---|---|---|---|
| CA | ME | AG | AUC | PG | AR@30 | AR@50 | AR@80 | AR@100 | $P^s,P^e$ | $P^{se}$ | $P^c$ | AUC | PG | AR@30 | AR@50 | AR@80 | AR@100 |
| | baseline-FI | | 67.51 | - | 66.54 | 71.02 | 74.47 | 75.86 | | baseline-TS | | 65.63 | - | 64.74 | 69.13 | 72.50 | 73.85 |
| ✓ | - | - | 67.91 | +0.40 | 67.11 | 71.40 | 74.81 | 76.24 | ✓ | - | - | 68.21 | +2.58 | 67.58 | 71.56 | 74.93 | 76.28 |
| ✓ | ✓ | - | 68.42 | +0.91 | 67.52 | 71.92 | 75.23 | 76.55 | ✓ | ✓ | - | 68.50 | +2.87 | 67.87 | 72.06 | 75.48 | 76.93 |
| - | - | ✓ | 68.47 | +0.96 | 67.64 | 72.03 | 75.57 | 76.91 | ✓ | - | ✓ | 68.60 | +2.97 | 67.77 | 72.34 | 75.37 | 76.73 |
| ✓ | ✓ | ✓ | **69.36** | +1.85 | **68.79** | **73.08** | **76.24** | **77.59** | ✓ | ✓ | ✓ | **69.36** | +3.73 | **68.79** | **73.08** | **76.24** | **77.59** |
| self-attention | ✓ | ✓ | 69.16 | +1.65 | 68.65 | 72.81 | 76.00 | 77.37 | - | - | - | - | - | - | - | - | - |
| ✓ | self-excitation | ✓ | 69.10 | +1.59 | 68.38 | 72.77 | 76.12 | 77.29 | - | - | - | - | - | - | - | - | - |

Table 3: Ablation study of transformer-driven scorers module on ActivityNet-1.3 validation split.

| Input | | Transformer-driven Scorers | | | | | | Input | | Transformer-driven Scorers | | | | | |
|---|---|---|---|---|---|---|---|---|---|---|---|---|---|---|---|
| Encoder | Decoder | AUC | PG | AR@30 | AR@50 | AR@80 | AR@100 | Encoder | Decoder | AUC | PG | AR@30 | AR@50 | AR@80 | AR@100 |
| $\tilde{A}\parallel\tilde{M}$ | $\tilde{A}\parallel\tilde{M}$ | 67.49 | - | 66.62 | 71.23 | 74.33 | 75.90 | **F** | $\tilde{A}\parallel\tilde{M}$ | 68.34 | +0.85 | 67.49 | 71.88 | 75.20 | 76.54 |
| **F** | **F** | 69.19 | +1.70 | 68.59 | 72.77 | 76.11 | 77.42 | $\tilde{A}\parallel\tilde{M}$ | **F** | 69.36 | +1.87 | **68.79** | **73.08** | **76.24** | **77.59** |

| Head Selection | | Transformer-driven Scorers | | | | | | Head Selection | | Transformer-driven Scorers | | | | | |
|---|---|---|---|---|---|---|---|---|---|---|---|---|---|---|---|
| # heads | Operation | AUC | PG | AR@30 | AR@50 | AR@80 | AR@100 | # heads | Operation | AUC | PG | AR@30 | AR@50 | AR@80 | AR@100 |
| 8 | Concatenation | 69.07 | - | 68.53 | 72.59 | 75.90 | 77.22 | 8 | Selection | 69.36 | +0.29 | **68.79** | **73.08** | **76.24** | **77.59** |

features over the channel dimension for the subsequent proposal generation. The comparisons show that all three components, *i.e.*, CA, ME, AG, contribute positively. Precisely, the CA & ME improve AUC by $0.91\%$ and the AG module improves AUC by $0.96\%$, which respectively shows that concerning the two stream's interaction and the multi-scale temporal contexts is beneficial for obtaining the robust video representation. With the complete FI module, we can further improve AUC by $1.85\%$. Furthermore, the bottom two rows show the comparison results once we employ self-adjustment on each single-stream feature. The results show that the two-stream interaction is indeed advantaged. Precisely, compared with our full model, $i$) when replacing co-attention with self-attention, the AUC declines to $69.16$ by $0.20\%$; $ii$) when replacing mutual-excitation with self-excitation, the AUC also declines to $69.10$ by $0.26\%$.

**Transformer-driven Scorers.** The right part in Table 2 compares the probability setting within the TS module. The first row, *i.e.*, baseline-TS, serves as the baseline that only employs the probabilities of $P^s$ & $P^e$ to generate proposals. The comparisons show that all these scoring probabilities contribute positively. Precisely, the $P^s$ & $P^e$ using our boundary co-estimation has significantly improved the AUC by $2.58\%$. With the aids of the proposal-level scoring probabilities $P^{se}$ and $P^c$, we can improve more AUC by $0.29\%$ and $0.39\%$. As a result, our complete TS module remarkably improves AUC by $3.73\%$. The ablation results demonstrate that it is beneficial to retrieve action proposals concerning the boundary co-estimation and the association of the self-attended representations. Our TS-based approach fulfills the goal with noticeable performance gain.

Table 3 ablates the configurations of the TS module. The top part shows the various inputs for the encoder-decoder pair. The ablation shows that though the aggregated representation **F** helps use in the encoder-decoder pair, simultaneous employing the $\tilde{A}\parallel\tilde{M}$ & **F** gains more performance improvement. The bottom part shows the advantage of our head selection. Comparing to concatenating multiple heads directly, our head selection shows higher performance improvement.

**Visualization.** Figure 3 visualizes the effects of employing our co-attention and mutual-excitation and transformer-driven scorers. The left figures select the two-stream features of the $70th$ snippet locating at an action instance for comparison. The result shows that our co-attention and mutual-excitation modules obviously reduces the feature discrepancies between two-stream features. As a result, our proposal generation can employ the two-stream features without noticeable preferences. The right four figures show the estimated action-boundary probabilities between "baseline-TS" and our "full model." In comparison with baseline-TS, the full model estimates the boundary probabilities by using probabilities $P^s$, $P^e$, $P^{se}$, and $P^c$. The results show that our model contributes to estimating less false positive estimations, which again demonstrate the effectiveness of our FITS model.

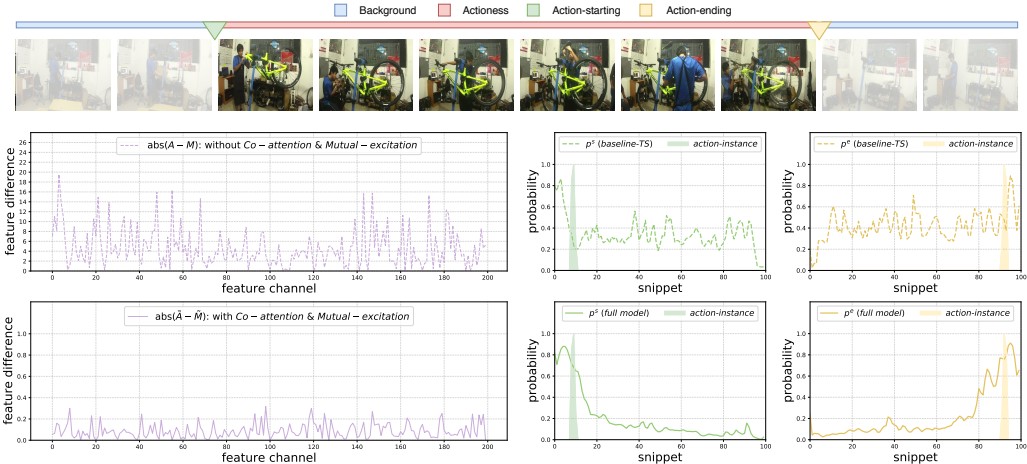

Figure 3: Effect visualization of our model on video id "pIk9qMEyEd4." The top row shows the sampled frames of the video. The left two figures in the bottom image set, which depict the absolute differences (y-axis) between motion cue and appearance cue over feature channels (x-axis), show our co-attention and mutual-excitation mechanism can reduce the two-stream feature discrepancies. The right four figures in the bottom image set, which plot the predicted boundary probabilities (y-axis) $P^s$ and $P^e$ over the snippet dimension (x-axis), show our TS module's effect. In this four-image set, the top row shows the original results, and the bottom row shows the results using our modules.

## 5 CONCLUSIONS

We have shown that the proposed FITS model, which is composed of the Feature-Integration module and the Transformer-driven Scorers module, better addresses the temporal action proposal generation task and achieves state-of-the-art performance. Specifically, the proposed action-boundary co-estimation mechanism shows its advantages in retrieving proposals of less false-positive predictions and, hence, helps retain high-quality proposals. The extensive experiments show that the performance gain is derived from not only the Feature-Integration module, which enriches and integrates the two-stream features as a robust snippet-level representation, but also the Transformer-driven Scorers module, which first generates the self-attended representations concerning the temporal contextual supports and then associates these self-attended representations for proposal scoring. As a result, our FITS model effectively utilizes the standard two-stream features and explores the self-attended video representations, therefore, lead to state-of-the-art TAPG performances on two challenging datasets.

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

## 6 APPENDIX

**Implementation Details.**   We use the two-stream features (Xiong et al., 2016) pre-trained on the training set of ActivityNet-1.3 with the same parameter settings as (Gao et al., 2017; Lin et al., 2018). We set 16 frames per snippet in ActivityNet-1.3 and 5 frames per snippet in THUMOS-14. Further, we sample the snippets with $T = 100$ via linear interpolation in ActivityNet-1.3 and with $T = 128$ via truncation and overlapped sliding windows in THUMOS-14. All the snippet manipulations are the same as (Lin et al., 2019; 2018). For the setting of soft-NMS, we respectively use threshold $0.8$ and $0.65$ for ActivityNet-1.3 and THUMOS-14, and the same decay parameter $0.85$ for both datasets. We train our model using Adam optimizer with batch size $24$ in ActivityNet-1.3 and batch size $4$ in THUMOS-14. The initial learning rate is $10^{-3}$ with decayed by $10^{-3}$ for every 10 epochs. The channel numbers $C = 200$, $C' = 1600$, and $C'' = 400$. The weighting factor $\lambda$ in (13) is set to $0.5$.

**Comparison against SOTA proposal generators**   Table 4 and Table 5 show the complete summarization of our proposal generator, *i.e.*, FITS, against state-of-the-art proposal generators on ActivityNet-1.3 and THUMOS-14, respectively. In both tables, the notation "*" indicates the method using non-two-stream features, and the notation "+" indicates the supplementary results. Our results consistently outperform all other state-of-the-art proposal generators on two datasets. Please note that the results of the ActivityNet-1.3 testing split are evaluated using the official server.

Table 4: Comparison of the state-of-the-art methods on ActivityNet-1.3 validation and testing splits.

| Method | +TCN Dai et al. (2017) | +MSRA Ting et al. (2017) | +Prop-SSAD Lin et al. (2017b) | +CTAP Gao et al. (2018) | FITS |
|---|---|---|---|---|---|
| AR@100 (val) | - | - | 73.01 | 73.17 | **77.59** |
| AUC (val) | 59.58 | 63.12 | 64.40 | 65.72 | **69.36** |
| AUC (test) | 61.56 | 64.18 | 64.80 | - | **70.00** |

| Method | +BSN (Lin et al., 2018) | +MGG (Liu et al., 2019) | BMN (Lin et al., 2019) | *GTAN (Long et al., 2019) | FITS |
|---|---|---|---|---|---|
| AR@100 (val) | 74.16 | 74.54 | 75.01 | 74.80 | **77.59** |
| AUC (val) | 66.17 | 66.43 | 67.10 | 67.10 | **69.36** |
| AUC (test) | 66.26 | 66.47 | 67.19 | 67.40 | **70.00** |

| Method | RapNet (Gao et al., 2020) | DBG Lin et al. (2020) | Zhao et al. (2020) | BC-GNN (Bai et al., 2020) | FITS |
|---|---|---|---|---|---|
| AR@100 (val) | 76.71 | 76.65 | 75.27 | 76.73 | **77.59** |
| AUC (val) | 67.63 | 68.23 | 66.51 | 68.05 | **69.36** |
| AUC (test) | 67.72 | 68.57 | - | - | **70.00** |

Table 5: Comparison of the state-of-the-art methods on THUMOS-14 testing split.

| Method | AR@50 | AR@100 | AR@200 | AR@500 | AR@1000 |
|---|---|---|---|---|---|
| +*SCNN-prop(Shou et al., 2016) | 17.22 | 26.17 | 37.01 | 51.57 | 58.20 |
| +*SST (Buch et al., 2017b) | 19.90 | 28.36 | 37.90 | 51.58 | 60.27 |
| +TAG (Zhao et al., 2017) | 18.55 | 29.00 | 39.61 | - | - |
| +TURN (Gao et al., 2017) | 21.86 | 31.89 | 43.02 | 57.63 | 64.17 |
| +CTAP (Gao et al., 2018) | 32.49 | 42.61 | 51.97 | - | - |
| +BSN (Lin et al., 2018) | 37.46 | 46.06 | 53.21 | 60.64 | 64.52 |
| +MGG (Liu et al., 2019) | 39.93 | 47.75 | 54.65 | 61.36 | 64.06 |
| *GTAN (Long et al., 2019) | - | - | 54.30 | - | - |
| BMN (Lin et al., 2019) | 39.36 | 47.72 | 54.70 | 62.07 | 65.49 |
| RapNet (Gao et al., 2020) | 40.35 | 48.23 | 54.92 | 61.41 | 64.47 |
| DBG (Lin et al., 2020) | 37.32 | 46.67 | 54.50 | 62.21 | 66.40 |
| Zhao et al. (2020) | **44.23** | **50.67** | 55.74 | - | - |
| BC-GNN (Bai et al., 2020) | 40.50 | 49.60 | 56.33 | 62.80 | 66.57 |
| FITS | 40.32 | 49.53 | **57.55** | **65.34** | **69.56** |

Table 6: Temporal action detection results on ActivityNet-1.3 validation split. Notation '*' indicates the model leveraging the results from UntrimmedNet-based action classifier (Wang et al., 2017).

| Method | mAP@0.5 | mAP@0.75 | mAP@0.95 | Average |
|---|---|---|---|---|
| CDC (Shou et al., 2017) | 43.83 | 25.88 | 0.21 | 22.77 |
| SSN (Zhao et al., 2017) | 39.12 | 23.48 | 5.49 | 23.98 |
| *BSN (Lin et al., 2018) | 46.45 | 29.96 | 8.02 | 30.03 |
| *P-GCN (Zeng et al., 2019) | 48.26 | 33.16 | 3.27 | 31.11 |
| *BMN (Lin et al., 2019) | 50.07 | 34.78 | 8.29 | 33.85 |
| *G-TAD (Xu et al., 2020) | 50.36 | 34.60 | 9.02 | 34.09 |
| *BC-GNN (Bai et al., 2020) | 50.56 | 34.75 | 9.37 | 34.26 |
| FITS | **51.89** | **35.05** | **10.16** | **34.75** |

**Action detection with FITS action proposals**    Due to the limited space allowed in the main paper, we provide the experimental results of temporal action detection task in the supplementary material. The methods (Buch et al., 2017b; Escorcia et al., 2016; Gao et al., 2018; 2020; 2017; 2019; Lin et al., 2020; 2019; 2018; Liu et al., 2019; Long et al., 2019; Shou et al., 2016; Xu et al., 2017; Zhao et al., 2017) have been mentioned in the main paper. For assessing the quality of our proposals for helping a video action classifier, we first feed FITS proposals into the state-of-the-art temporal action classifier, *i.e.*, P-GCN (Zeng et al., 2019) [1], in which the P-GCN classifier infers the action probability distribution of each proposal. We then re-score each proposal by multiplying its action score, the highest specific-action probability derived from the P-GCN classifier, with the FITS proposal score, the equation (12) in the main paper. Finally, we re-rank the proposals according to their multiplied scores for evaluation.

Table 6 shows the detection performance with top-100 proposals in metric mAP@IoU, which evaluates the relation between mean Average Precision (mAP) and specified IoU thresholds, compared to the state-of-the-art methods on ActivityNet-1.3 validation split. Following the methods (Lin et al., 2018; Zeng et al., 2019; Xu et al., 2020), we adopt the video-level action probability distribution derived from the UntrimmedNet-based action classifier (Wang et al., 2017). We then multiply the action probability to our FITS proposal score, *i.e.*, the equation (12) defined in the main paper.

Table 7 shows the detection performance in metric mAP@IoU, which evaluates the relation between mean Average Precision (mAP) and specified IoU thresholds, compared to the state-of-the-art methods on THUMOS-14 testing split. Note that the original P-GCN method (Zeng et al., 2019) adopts the proposals generated by BSN Lin et al. (2018). When replacing the BSN proposals of P-GCN with our FITS proposals, our detection result "FITS+P-GCN" achieves 5% performance gain compared to P-GCN, 4% performance gain compared to "Zhao et al. (2020)+P-GCN,", and 2.5% performance gain compared to "G-TAD+P-GCN," respectively. The experiment shows the advantage of our proposals to improve an action classifier achieving state-of-the-art performance while addressing the action detection task.

---

[1]P-GCN implementation: https://github.com/Alvin-Zeng/PGCN

Table 7: Temporal action detection results on THUMOS-14 testing split.

| Method | mAP@0.1 | mAP@0.2 | mAP@0.3 | mAP@0.4 | mAP@0.5 | mAP@0.6 | mAP@0.7 |
|---|---|---|---|---|---|---|---|
| DAP (Escorcia et al., 2016) | - | - | - | - | 13.9 | - | - |
| S-CNN (Shou et al., 2016) | 47.7 | 43.5 | 36.3 | 28.7 | 19.0 | - | - |
| SSAD (Lin et al., 2017a) | 50.1 | 47.8 | 43.0 | 35.0 | 24.6 | - | - |
| SS-TAD (Buch et al., 2017a) | - | - | 45.7 | - | 29.2 | - | 9.6 |
| SST (Buch et al., 2017b) | - | - | 37.8 | - | 23.0 | - | - |
| CDC (Shou et al., 2017) | - | - | 40.1 | 29.4 | 23.3 | 13.1 | 7.9 |
| TURN (Gao et al., 2017) | 54.0 | 50.9 | 44.1 | 34.9 | 25.6 | - | - |
| R-C3D (Xu et al., 2017) | 54.5 | 51.5 | 44.8 | 35.6 | 28.9 | - | - |
| SSN (Zhao et al., 2017) | 66.0 | 59.4 | 51.9 | 41.0 | 29.8 | - | - |
| ETP (Qiu et al., 2018) | - | - | 48.2 | 42.4 | 34.2 | 23.4 | 13.9 |
| CTAP (Gao et al., 2018) | - | - | - | - | 29.9 | - | - |
| BSN (Lin et al., 2018) | - | - | 53.5 | 45.0 | 36.9 | 28.4 | 20.0 |
| TAL-Net (Chao et al., 2018) | 59.8 | 57.1 | 53.2 | 48.5 | 42.8 | 33.8 | 20.8 |
| MGG (Liu et al., 2019) | - | - | 53.9 | 46.8 | 37.4 | 29.5 | 21.3 |
| GTAN (Long et al., 2019) | 69.1 | 63.7 | 57.8 | 47.2 | 38.8 | - | - |
| BMN (Lin et al., 2019) | - | - | 56.0 | 47.4 | 38.8 | 29.7 | 20.5 |
| P-GCN (Zeng et al., 2019) | 69.5 | 67.8 | 63.6 | 57.8 | 49.1 | - | - |
| DBS (Gao et al., 2019) | 56.7 | 54.7 | 50.6 | 43.1 | 34.3 | 24.4 | 14.7 |
| DBG (Lin et al., 2020) | - | - | 57.8 | 49.4 | 39.8 | 30.2 | 21.7 |
| FC-AGCN-P-C3D (Li et al., 2020) | 59.3 | 59.6 | 57.1 | 51.6 | 38.6 | 28.9 | 17.0 |
| PBRNet (Liu & Wang, 2020) | - | - | 58.5 | 54.6 | 51.3 | **41.8** | **29.5** |
| G-TAD (Xu et al., 2020) | - | - | 54.5 | 47.6 | 40.2 | 30.8 | 23.4 |
| Zhao et al. (2020) | - | - | 53.9 | 50.7 | 45.4 | 38.0 | 28.5 |
| BC-GNN (Bai et al., 2020) | - | - | 57.1 | 49.1 | 40.4 | 31.2 | 23.1 |
| G-TAD (Xu et al., 2020)+P-GCN | - | - | 66.4 | 60.4 | 51.6 | 37.6 | 22.9 |
| Zhao et al. (2020)+P-GCN | 71.8 | 70.3 | 66.3 | 61.0 | 50.1 | - | - |
| FITS+P-GCN | **75.0** | **72.7** | **69.1** | **62.7** | **54.1** | 41.1 | 27.8 |

