# OpenReview forum: "Feature Integration and Group Transformers for Action Proposal Generation"
_ICLR.cc/2021/Conference — Reject_

### Official Review · AnonReviewer4 · 2020-10-26

**Rating:** 5
**Confidence:** 3

**Review:**

Summary:

This paper proposed two modules for temporal action proposal generation in videos: feature integration and group transformers. Both modules rely on self-attention and transformer architectures to improve the feature representation and the scoring component. Experiments on two well-known temporal action datasets shows the potential of the proposed approach.

Strengths:

The paper is well written and nicely visualized. The novelties of the paper are clearly outlined, while the modules are carefully presented in Section 3 and in Figures 1 and 2. This presentation makes the paper easy to understand and to build upon.

The experiments are accompanied with multiple ablation studies detailing the effect of different components. The studies show which parts are most effective and how they interact.

Weaknesses:

[Side note] While the paper does directly deal with learning representations, the topic of temporal action proposal generation seems like a much better fit for a computer vision conference. The relevant papers cited are also from CVPR/ICCV/ECCV/etc. While no grounds for a negative rating here, it will likely negatively affect the impact of the paper to not submit it to a computer vision conference.

On the paper itself, a core limitation is that the innovation of the paper is on the marginal side. The two novelties: feature integration and transformer-driven scorers, consist of existing components added to the temporal action proposal pipeline. The feature integration makes use of attention (through co-attention), akin to the encoder. The transformer-driven are deemed the most innovate part of the paper, although at its core, the transformers mainly attend for representations. Overall, it is interesting to see that attention and transformers can be applied to the problem of temporal action proposal generation, but the insights gained from the approach are modest.

The experimentation contains a number of limitations to be addressed:
- The qualitative visualization in its current form provides no insight. First, it is unclear to the reader what the video is, only the id "pIk9qMEyEd4" is provided. Furthermore, the graphs are not clear. Why is reducing the two-stream feature discrepancy important for the mysterious video? What is action-instance? Is green the start and yellow the end?

- The reviewer appreciates that the state-of-the-art comparisons include references to ECCV 2020, which took place only a little over a month before the ICLR deadline. The comparison to (Bai et al., 2020) does raise a number of questions. First, it seems that the submission performs worse for low average numbers of proposals, and better for high average numbers of proposals. In general, action proposal generation is not the end goal, but the first step for the task of temporal action detection. The question is: what is more effective for action detection? Better AR@50 or better AR@500? (Bai et al., 2020) include action detection results for their approach and it would be good to have the same for the proposal approach.

- What is missing in the paper is insights into what makes the two modules effective for the task at hand. The ablation studies show that the components help to improve the metric, but why are the modules effective? What problem do they solve? Perhaps success and failure cases can help with understanding the benefit of the modules.

Conclusion:

Overall, the paper is well written and clearly presented. The ability to integrate attention and transformers is an interesting but modest addition to the problem of temporal action proposal generation. The experiments contain good ablation studies, but also have a number of open questions. For these reasons, a current rating hovering borderline is given. For the rebuttal, it would be interesting to see discussions regarding the novelty of the approach and the questions raised in the experiments.

Opinion after rebuttal:

Originally, my rating was a 5. The authors have rebutted my concerns regarding the experimentation, which have gained in insight. The limited novelty and fit to the conference do remain pressing issues and it seems that this is partially shared with the other reviewers. My recommendation is therefore to extend and resubmit.

---

> ### Author Response · Authors · 2020-11-25
> **Responses to AnonReviewer4**
>
> 4.1 “The qualitative visualization in its current form provides no insight. First, it is unclear to the reader what the video is, only the id "pIk9qMEyEd4" is provided. Furthermore, the graphs are not clear. Why is reducing the two-stream feature discrepancy important for the mysterious video? What is action-instance? Is green the start and yellow the end?”
>
> Thank you for the feedback. We have attached some frames in Figure 3 for visualization. The two-stream feature is designed to describe the action-probability among the common 200 action-classes, which corresponds to the feature-channel-axis of the left sub-figure in Figure 3, from the aspects of appearance and motion. That is, the higher two-stream feature discrepancy implies that the action predictions of the two aspects are of higher inconsistency. Therefore, we assume that reducing the two-stream feature discrepancy is able to reduce the ambiguity while discriminating the video actions. The ablation result in Table 2 supports our assumption. The action instance represents a ground-truth action in a video, and we respectively denote its start in green and the end in yellow in Figure 3.
>
>
> 4.2 ”The reviewer appreciates that the state-of-the-art comparisons include references to ECCV 2020, which took place only a little over a month before the ICLR deadline. The comparison to (Bai et al., 2020) does raise a number of questions. First, it seems that the submission performs worse for low average numbers of proposals, and better for high average numbers of proposals. In general, action proposal generation is not the end goal, but the first step for the task of temporal action detection. The question is: what is more effective for action detection? Better AR@50 or better AR@500? (Bai et al., 2020) include action detection results for their approach and it would be good to have the same for the proposal approach.”
>
> Thank you for the suggestion. We have carried out the action detection experiment by integrating our method for proposal generation with an action classifier. In the appendix of our revised paper, both Table 6 and Table 7 show that our action detection performance is better than (Bai et al., 2020). However, “better AR@50” or “better AR@500” is more effective for action detection is still a trad-off problem since “better AR@50” may imply the low action-recall and “better AR@500” may imply the low action-accuracy.
>
>
> 4.3 ”What is missing in the paper is insights into what makes the two modules effective for the task at hand. The ablation studies show that the components help to improve the metric, but why are the modules effective? What problem do they solve? Perhaps success and failure cases can help with understanding the benefit of the modules.”
>
> Thank you for the feedback. As we indicate in the abstract, most previous methods estimate action boundaries without considering the long-range temporal neighborhood. The proposed action-boundary co-estimation mechanism in the TS module leverages the bi-directional contextual supports to facilitate such boundary estimation, which shows the advantage of removing several false-positive boundary predictions. To adopt the TS module, we found that directly applying the two-stream feature is not adequate due to the higher two-stream feature discrepancy. Hence, we design the FI module to seamlessly fuse two-stream features concerning their interaction to reduce their feature discrepancy. The experimental results support our assumption and the proposed implementation is helpful to tackle the TAPG task.

---

### Official Review · AnonReviewer2 · 2020-10-28
**An interesting application paper that models long-term temporal feature for action proposal generation**

**Rating:** 6
**Confidence:** 4

**Review:**

- The paper proposed a feature integration (FI) module and utilized transformers to capture long-range temporal dependencies, which is reasonable and interesting. However, these two modules focus on different aspects for action proposal generation. Is there a connection between these two modules?

- The proposed method was evaluated on two popular benchmarks, and achieved convincing results. Besides this, a comprehensive ablation study is provided, which is helpful to understand the contribution of each part.

- For the transformer, each of them uses the A||M as input for encoder, but F as input for decoder. F is a feature vector computed from A and M, is there any intuition about why A||M are used for encoder and F for decoder?

- More qualitative results and analysis should be provided. For example, the proposed method cannot beat the performance of Zhao et al (2020) on THUMOS-14, and is there any result or analysis could explain this?

- For implementation details, details about transformer, such as number encoder and decoder layers, are not provided, which could make the re-implementation of the paper as a concern.

Overall, it is an interesting paper, and I lean to borderline accept before the rebuttal.

---

> ### Author Response · Authors · 2020-11-25
> **Responses to AnonReviewer2**
>
> 2.1: "Is there a connection between these two modules?"
>
> Thank you for the feedback. The proposed FITS model first employs the Feature-Integration module to locally enrich and integrate the two-stream features. Then it uses the Transformer-driven Scorers module to consider the long-range temporal neighborhood for generating the self-attended representations and scoring proposals.
>
> 2.2: "For the transformer, each of them uses the A||M as input for encoder, but F as input for decoder. F is a feature vector computed from A and M, is there any intuition about why A||M are used for encoder and F for decoder?"
>
> Thank you for the feedback. In the FI module, F is akin to more informative representation of ~A || ~M.
> Within a Transformer, the decoder plays the role of a guider to conduct the representation learning of the encoder. While feeding the same feature into the Transformer, its attention mechanism would make analogies to self-attention and tends not to distill more information, as shown in Table 3. Hence, our model employs a high-level representation F to guide the representation learning of a low-level representation A || M. The comparison results in Table 3 support the proposed setting.
>
> 2.3: "More qualitative results and analysis should be provided. For example, the proposed method cannot beat the performance of Zhao et al (2020) on THUMOS-14, and is there any result or analysis could explain this?"
>
> From the results in Tables 1, 5 and 7, the method by (Zhao et al., 2020) has better performance than ours on THUMOS-14 when generating smaller numbers of proposals. The results suggest that the proposal ranking of the method (Zhao et al., 2020) is effective while dealing with the THUMOS-14 dataset. However, for a practical proposal generator, the ability to retain most of the potential action proposals is important for the subsequent video analysis task. To this aim, the proposed FITS model achieves a better result while tackling the action detection task. Please refer to Tables 6 and 7 for comparing the action detection performance.
>
> 2.4: "For implementation details, details about transformer, such as number encoder and decoder layers, are not provided, which could make the re-implementation of the paper as a concern"
>
> Thank you for the feedback. We adopt a one-layer eight-head Transformer, as described in the first paragraph of Section 3.2. We will make our code publicly available for further contributions.

---

### Official Review · AnonReviewer1 · 2020-10-29
**Good motivation but kind of tricky and engineering**

**Rating:** 5
**Confidence:** 3

**Review:**

This paper tackles the problem of temporal action proposal generation (TAPG). The authors address the problem from two perspectives: features wise and score fusion wise. They use non-local blocks to integrate appearance features and motion features together. For score fusion, they propose transformer based module to incorporate long range temporal information. The proposed method is evaluated on two benchmark datasets and achieved state-of-the art performance.

Strength:
+ The paper is well written and easy to follow.
+ The analysis and ablation studies are thorough and adequate.

Weakness:
+ The novelty of the propose method is incremental, specifically:
(1). The Feature Integration module is standard  non-local blocks which have been used in many video related applications. Such feature integration module does not specific designed for action proposal generation. Even though the final performance benefits from the feature integration module, it's because of the improvement of feature representations. If you replace the two stream feature with more advanced video features, the performance will likely improve.
(2). Compared to previous scorer, the Transformer-driven Scorer uses the transformer backbone and outputs four additional scores for actioness and background. The difference between these scorers are not significant.

+ State-of-the-art performance is not necessary. These two papers that seems to have higher numbers though:
[a] Accurate Temporal Action Proposal Generation with Relation-Aware Pyramid Network, AAAI 2020.
[b] Sequence-to-Segments Networks for Detecting Segments in Videos, PAMI 2019.

Questions:
1. Some details regarding how to train P^{fa}, P^{ba}, P^{fb}, P^{bb} are missing. What is the ground truth for these scores? My understanding is that the ground truth for P^{fa} and P^{ba} are mostly the same except for the boundary?
2. Why separate transformer modules are needed for backward and forward? This increased a lot of extra parameters. Unlike RNNs, either feeding the stream forward or backward doesn't matter for transformers. Why not share the same transformer for both forward and backward? Only changes the final FC layers for backward and forward?
3. The conclusion from Table 3 top part is confusing. Why using integrated feature F for both encoders and decoders does not perform the best?


Summary:
My major concern is about novelty. Although it has good performance, this paper seems kinda tricky and engineering.


---------
Update after rebuttal:

The updated paper addressed my concerns regarding missing details. I appreciated the efforts of comparing with more sophisticated backbone features. However, as pointed by other reviewers, the novelty of the paper is marginal. I keep my original rating as 5.

---

> ### Author Response · Authors · 2020-11-25
> **Responses to AnonReviewer1**
>
> 1.1: "The novelty of the propose method is incremental, specifically: (1). The Feature Integration module is standard non-local blocks which have been used in many video related applications. Such feature integration module does not specific designed for action proposal generation. Even though the final performance benefits from the feature integration module, it's because of the improvement of feature representations. If you replace the two stream feature with more advanced video features, the performance will likely improve. (2). Compared to previous scorer, the Transformer-driven Scorer uses the transformer backbone and outputs four additional scores for actioness and background. The difference between these scorers are not significant."
>
> Thank you for the feedback. (1) The proposed FI module is designed to better explore the mutual information of the two-stream features. As shown in section 3.1, the design does not simply utilize off-the-shelf functions, and the proposed mutual-excitation scheme is new. Although our formulation does not employ more advanced video features, the FI module does help achieve state-of-the-art performance. (2) We believe that the introduction of the transformer-driven scorers are significant. In particular, they can be coupled to improve the prediction of action boundaries. Take, for example, that in the estimation of the probability of action starting P^s in equation (9), We can conveniently concatenate F^{ba} with F^{fb} to facilitate the prediction. This is based on the property that an action starting boundary is more likely to occur if its time neighborhood includes high responses of backward-actioness and forward-background. An analogous coupling can also be used to improve the prediction of the probability of action ending P^e as in (10).
>
> 1.2: "State-of-the-art performance is not necessary. These two papers that seems to have higher numbers though: [a] Accurate Temporal Action Proposal Generation with Relation-Aware Pyramid Network, AAAI 2020. [b] Sequence-to-Segments Networks for Detecting Segments in Videos, PAMI 2019."
>
> The slightly better performances of the two mentioned techniques are owing to their use of a more sophisticated backbone for feature generation. We have experimented with the same backbone used in the AAAI 2020 work, our approach improves the AUC on the validation split and testing split from 67.63 and 67.72 to 69.36 and 70.0, respectively. We will including these findings in the final version of our work.
>
> 1.3: "Some details regarding how to train P^{fa}, P^{ba}, P^{fb}, P^{bb} are missing. What is the ground truth for these scores? My understanding is that the ground truth for P^{fa} and P^{ba} are mostly the same except for the boundary?"
>
> Thank you for the feedback. We have described the ground-truth label assignment in the section of experiments in the revised paper. Please refer to the label assignment in that section for more details.
>
> 1.4: "Why separate transformer modules are needed for backward and forward? This increased a lot of extra parameters. Unlike RNNs, either feeding the stream forward or backward doesn't matter for transformers. Why not share the same transformer for both forward and backward? Only changes the final FC layers for backward and forward?"
>
> Transformer is a sequence-to-sequence model and the order of information accumulation does matter. The forward-order transformer yields forward-actioness and forward-background, while the backward-order transformer gives backward-actioness and backward-background. We emphasize again that these four scorers and their couplings in equations (9) and (10) are pivotal in our formulation to better predict the action boundaries.
>
> 1.5: "The conclusion from Table 3 top part is confusing. Why using integrated feature F for both encoders and decoders does not perform the best?"
>
> Within a Transformer, the decoder plays the role of a guider to conduct the representation learning of the encoder. While feeding the same feature into the Transformer, its attention mechanism would make analogies to self-attention and tends not to distill more information, as shown in Table 3. Hence, our model employs a high-level representation F to guide the representation learning of a low-level representation A || M. The comparison results in Table 3 support the proposed setting.

---

### Official Review · AnonReviewer3 · 2020-10-30
**This paper targets two issues in action proposal generation: feature representation and scoring, and provide a feature integration module and transformer-driven scorer respectively. The performance on two datasets is promising.**

**Rating:** 5
**Confidence:** 5

**Review:**

In general, it is an interesting paper to utilize multiple techniques to enhance two-stream features and transformer to improve proposal scores, though all the techniques are not first proposed in this paper. But some technical details are not clearly presented, so the solidarity cannot be evaluated. Furturemore, more ablation study is needed to verify the contribution of the paper. If all the the concerns are properly addressed in the rebuttal, this paper can be accepted.

- Quality: The overall quality is ok but not good enough considering that the idea is interesting and the performance is promising, but the presentation is not quite clear.
- Clarity: This needs improvement. Please see the following detailed comments.
- Originality: All techniques used in this paper are not originally proposed, but the ways they are utilized for this specific problem have novelty.
- Significance: This work has promising performance on action proposals, but its significance can only be evaluated after the detection performance is provided.

Detailed comments:
1) It is not mentioned how the groundtruth labels for the 8 different probabilities in Eq. (13) are generated, especially for p^c and p^{se}. [Addressed]
2)  It is not clear how the forward and backward transformers are computed. It is mentioned at Page 5 that the input for backward are reversed. Does it mean that the first snippet becomes the last one in the backward transformer? If so, how can F^{ba} and F^{fb} be directly concatenated considering that their snippets indices do not match?  How does a backward transformer differ from a forward transformer considering that self-attention is not uni-directional? [Addressed]
3) The feature integration part is claimed to utilize the interaction of the two streams to fuse them with co-attention and mutual-excitation. Does co-attention work better than self-attention on each stream? Does mutual-excitation work bettern than self-excitation on each stream? Ablation experiments should be provided to verify that the two stream interaction is indeed important.  [According to the response, the advantage of co-attention and mutual-excitation seems marginal. That makes the claimed novelty of proposing these modules trivial because attention and self-excitation are existing works. ]
4) Though the paper claims to solve the problem of action proposal generation, the purpose of generating action proposals is to do action detection. So action detection performance is expected on the two datasets as well.  [Addressed]
5) Minor issues.
- In the 'Mutual-excitation' part on Page 4, it says that 1*1 convolutional filter is used on the C*T feature. What does that mean? How could you apply 1*1 convolution on a two-dimensional feature? Does it actually compute anything?
- In the 'Aggeration' part, the convolutional kernel for the temporal convolution is written as '1*3', which should be '3' because you only have 1 dimention apart from the feature dimension.
- In the 'Aggeration' part, the purpose is claimed to take advantage of 'multi-scale temporal contexts'. From Eq. (5), all the 4 features can be directly concatenated, which means that they have the same temporal size. So in Eq. (4), there is no stride for the max-pooling operation. Then why is it multi-scale?

--------------------------------
After rebuttal:
Though the paper is interesting in showing improved proposal and detection performance by using attention, excitation, and transformers, the novelty is not significant, especially when it is shown that the claimed cross-modality attention and excitation do not help much. The other reviewers also show the same concerns. I incline to degrade my score to 5.

---

> ### Author Response · Authors · 2020-11-25
> **Responses to AnonReviewer3**
>
> 1. We have detailed the ground-truth label assignment in the section of experiments in the revised paper. Please refer to the label assignment in that section for more details.
>
> 2. Thank you for the feedback.  (1) Yes, we reverse the input over temporal dimension to obtain a backward one. It implies that the first snippet becomes the last one in the backward transformer. However, before concatenating the two features of the two different directions, the backward ones will be reversed over the temporal dimension again. Therefore, the i-th snippet of the concatenated F^{ba} and F^{fb} is guaranteed to accumulate the backward information after i-th snippet from F^{ba} and the forward information before i-th snippet from F^{fb}. (2) Recall that the Transformer is a sequence-to-sequence model, which implies the generated representation is learned concerning the feature feeding order. Hence, the two transformers trained by the different feeding order are able to discriminate the actions of different playback directions.
> Owing to the mechanism of masked multi-head attention in decoder of transformer, we can obtain two distinct representations from bi-directional inputs.
>
> 3. To take account of the reviewer's feedback, we conduct more ablation experiments and show the comparison results in Table 2 of the revised paper. The results show that the two stream interaction is advantageous. Specifically, (1) when replacing co-attention with self-attention, the AUC declines to 69.16 by 0.20% with respect to our full model (AUC: 69.36); (2) when replacing mutual-excitation with self-excitation, the AUC declines to 69.10 by 0.26% in comparison with our full model (AUC: 69.36).
>
> 4. Thank you for the suggestion. We have carried out the action detection experiment by integrating our method for proposal generation with an action classifier. In the appendix of our revised paper, Table 6 shows the detection performance with top-100 proposals for evaluation in metric mAP@IoU on ActivityNet-1.3 validation split. The results show that the high-quality proposals from our FITS model improve action detection task.
>
> 5-1. "In the 'Mutual-excitation' part on Page 4, it says that 11 convolutional filter is used on the CT feature. What does that mean? How could you apply 1*1 convolution on a two-dimensional feature? Does it actually compute anything?"
>
> In our formulation, each snippet is encoded as one C-dimensional feature vector, which is treated as a 3D tensor of size C * 1 * 1. Hence we are able to employ the 1 * 1 convolution to project each snippet to a different channel size.
>
> 5-2. "In the 'Aggeration' part, the convolutional kernel for the temporal convolution is written as '1*3', which should be '3' because you only have 1 dimention apart from the feature dimension."
>
> We represent the convolutional kernel with respect to the input tensor's dimension to prevent the readers from guessing which dimensions the kernel is applied to. In the revised paper, we have modified the writing for the kernel representation.
>
> 5-3. "In the 'Aggeration' part, the purpose is claimed to take advantage of 'multi-scale temporal contexts'. From Eq. (5), all the 4 features can be directly concatenated, which means that they have the same temporal size. So in Eq. (4), there is no stride for the max-pooling operation. Then why is it multi-scale?"
>
> The operation of one 5 * 5 convolution is analogous to two successive layers of 3 * 3 convolution (Szegedy et al., 2016). In Eq. (5), the two features are derived from successive pooling and convolution, which implicitly consider a larger temporal context than the features derived from one single convolution. Hence we treat Eq. (5) as one kind of multi-scale aggregation.

---

### Decision · Program_Chairs · 2021-01-07
**Final Decision**

**Decision:**

Reject

**Comment:**

The paper focuses on the task of finding higher fidelity action proposals for temporal action proposal detection. As the reviewers mentioned, this task is a pre-task to temporal activity localization/detection in video, which is the main task to be solved. The paper may be perceived differently if it were presented as a detection method instead. Apart from the scope of the paper, the reviewers also unanimously agree on the limited technical novelty of the proposed methodology in the paper. The proposed method can be seen as an application of self-attention and transformer techniques on the problem of activity detection. The goal of these techniques is feature enrichment that serves to incorporate information across long-term context, a concept that has appeared previously in other work but not necessarily with the same machinery (e.g. G-TAD).

Despite its shortcomings and since it presents promising experimental results on well-known proposal/detection benchmarks, the authors can benefit from considering the reviewers' comments and suggestions to produce a stronger and more compelling future submission.